# *Cnidium monnieri* (L.) Cusson Flower as a Supplementary Food Promoting the Development and Reproduction of Ladybeetles *Harmonia axyridis* (Pallas) (Coleoptera: Coccinellidae)

**DOI:** 10.3390/plants12091786

**Published:** 2023-04-27

**Authors:** Wenwen Su, Fang Ouyang, Zhuo Li, Yiyang Yuan, Quanfeng Yang, Feng Ge

**Affiliations:** 1State Key Laboratory of Integrated Management of Pest Insects and Rodents, Institute of Zoology, Chinese Academy of Sciences, Beijing 100101, China; suwenwen@ioz.ac.cn (W.S.);; 2CAS Center for Excellence in Biotic Interactions, University of the Chinese Academy of Sciences, Beijing 100101, China; 3Institute of Plant Protection, Shandong Academy of Agricultural Sciences, Jinan 250100, China; 4State Key Laboratory of Urban and Regional Ecology, Research Center for Eco-Environmental Sciences, Chinese Academy of Sciences, Beijing 100085, China

**Keywords:** coccinellidae, supplementary food feeding, floral resource, development, reproduction

## Abstract

Predaceous ladybeetles are highly polyphagous predators that ingest supplementary food from flowering plants. Flowering plants widely grown in agroecosystems can sustain multiple natural enemies of agricultural pests, and the pollen and nectar resources from flowering plants may have a positive role in natural enemies. *Cnidium monnieri* (L.) Cusson, an annual herb with many flowers, blooms from May to July. *C. monnieri* can support several predatory natural enemies, and the addition of *C. monnieri* strips increases the density of *Harmonia axyridis* (Pallas) and improves the biological control of apple aphids in an apple orchard. *H. axyridis* is also the most important natural enemy in wheat aphid biocontrol and is attracted to healthy and aphid-infested *C. monnieri* plants. In addition, adult *Propylaea japonica* Thunberg survives significantly longer on *C. monnieri* flowers than on a water-only diet. In this study, a laboratory experiment was conducted to assess (i) the effect of nutritional supplements derived from *C. monnieri* flowers on the development and reproduction of *H. axyridis* under a wheat aphids-only diet; (ii) the effect of *C. monnieri* flowers on *H. axyridis* adult reproduction performance. We compared the larval durations, survival, weight, adult longevity, and reproduction of *H. axyridis* reared on wheat aphids-only and aphids plus *C. monnieri* flower diets. The results showed that *H. axyridis* larvae reared on aphids plus flowers had significantly greater weights and survival rates, shorter larval durations, and produced 1.62 times more eggs than those reared on wheat aphids-only diets. *H. axyridis* adults ingesting a *C. monnieri* flowers plus an aphid diet increased egg production 1.44 times compared to the aphids-only diet. Our study demonstrates that *C. monnieri* flowers as a supplementary food positively affect the survival, development, and reproduction performance of *H. axyridis.*

## 1. Introduction

The multicolored Asian ladybeetle *Harmonia axyridis* (Pallas) (Coleoptera: Coccinellidae) is a species native to central and eastern Asia and has been established in America, Europe, and Africa [1,2]. *H. axyridis* is a highly polyphagous predator that consumes mainly aphids but can also feed on Tetranychidae, Psyllidae, Coccoidea, Chrysomelidae, Curculionidae, and Lepidoptera. It has been recognized as a biological control agent for several species of aphid and coccid pests and has provided biocontrol services in various agroecosystems (apple orchards, citrus orchards, and red pines) [3,4]. In northern China, *H. axyridis* and *Propylaea japonica* Thunberg are the most important natural enemies for cereal aphids and cotton aphid biocontrol [5,6]. *H. axyridis* adult ingests nectar from wild flowers as well as extrafloral nectar [7]. Gut content analyses have shown that *H. axyridis* adults frequently visit and feed on floral resources such as pollen in fields [7,8]. Lundgren indicated that coccinellids adults use pollen and nectar from floral resources to increase survival when prey is scarce, reduce mortality during diapause, supply energy for migration, and enhance reproductive capacity. Thus, flowering plants may have a positive role in the sustainable use of natural enemies, such as the predator ladybeetle *H. axyridis*.

Likewise, in agroecosystems, the availability of floral resources is crucial for natural enemies [9]. Flowering plants are widely grown in non-crop habitats in agroecosystems and can sustain a diversity of natural enemies, including parasitoids and predators [10]. In highly intensive agricultural production systems, the loss of non-crop habitats leads to simplified agricultural landscapes and creates unsuitable conditions for natural enemies [11]. Manipulating non-crop habitats in agroecosystems to provide floral resources, alternative prey, and shelter for natural enemies is a method used in conservation biological control [10,12]. Designing flower strips as non-crop habitats in agroecosystems can sustain generalist predators and promote biocontrol [13,14].

The effects of flowering plants on natural enemies vary greatly among plant species. Thus, selecting suitable flowering plants to target natural enemies is crucial to improving their fitness and enhancing their biological control capacity [15]. For example, feeding on *Perilla frutescens* (Lamiaceae) flowers increased the longevity and fecundity of *H. axyridis* adults [16]. In parallel, clover pollen mixed with *Ephestia kuehniella* (Lepidoptera: Phycitidae) eggs significantly improved the longevity and reproductive performance of *H. axyridis* adults [17]. *Vicia sativa* (Fabaceae), *Fagopyrum esculentum* (Polygonaceae), and *Coriandrum sativum* (Apiaceae) improved the survival and reproduction of *H. axyridis* adults [18]. *Brassica* and *Sonchus* flowers, as supplementary foods (pollen and nectar), can improve the reproductive performance of *Hippodamia variegata*, specifically under conditions of prey limitation [19]. 

By contrast, Mauricio Silva de Lima revealed that pollen did not provide suitable nutrients for ovarian development in *Brumoides foudrasii* (Coleoptera: Coccinellidae) and appeared to exert an inhibitory effect on oviposition behavior [20]. Michaud and Grant demonstrated that the pollen of the cultivated sunflower, *Helianthus annus* L., proved fatal to *H. axyridis* larvae and adults [21]. *H. axyridis* reared on *Ageratum conyzoides* or *Bidens pilosa* in combination with *Anagasta kuehniella* (Zeller) (Lepidoptera: Pyralidae) had no oviposition [22]. Flowering plants allow predators to optimize their fitness by exploiting food sources, and simultaneously, some arthropod pest species also consume nectar and pollen [23], and floral resources can increase their longevity and oviposition [17]. Therefore, selecting suitable flowering plants to target natural enemies is important to improve their fitness, enhance their biological control capacity, and highlight the importance of selecting the most appropriate flowering plant species in or around agricultural fields.

*Cnidium monnieri* (L.) Cusson (Apiaceae: Cnidium) is a traditional Chinese herbal medicine that has been grown and used in China, Japan, Korea, and Vietnam. In China, it is widely grown in Hebei, Jiangsu, Zhejiang, Shandong, and Sichuan provinces [24]. *C. monnieri* naturally regrows from seeds in fields and roadsides and requires no particular care. In agricultural fields, it is sown in rows for seed harvesting. Our previous study found that *C. monnieri* is characterized by early blossom, long inflorescence, and high pollen yields and produces prolific small white flowers with shallow corollas that make pollen more easily foraged by natural enemies. It can sustain several predatory natural enemies, such as *H. axyridis* and *P. japonica*, during its flowering period [25,26]. In an apple orchard experiment, planting *C. monnieri* significantly increased the density of predators (*H. axyridis*, *Chrysoperla sinica*, *Episyrphus balteata*) on apple trees and significantly decreased the density of spirea aphids [27]. In a field experiment, planting *C. monnieri* flower strips at the border of wheat-maize rotation fields served as a bridge habitat to conserve ladybeetles in wheat fields during harvest and helped the predator to immigrate to adjacent maize fields for pest control [9].

Furthermore, a Y-tube olfactometer laboratory experiment indicated that *H. axyridis* adults were attracted to healthy and aphid-infested *C. monnieri* plants. In addition, *H. axyridis* adults were attracted to 1,2-diethylbenzene and p-diethylbenzene synthetic volatile compounds detected among the volatile blends emanating from healthy and aphid-infested plants in the GC–MS and GC-EAD trials [27]. A laboratory experiment indicated that *C. monnieri* as a floral food for *P. japonica* when prey is insufficient could significantly affect the longevity of *P. japonica* adults compared to a water-only diet [9]. However, to date, the effect of *C. monnieri* flowers as a supplementary food on larval development and reproduction of *H. axyridis* adults has not been studied.

Previous studies have focused on the effects of pollen and sugar diets or their mixed diets with prey on ladybeetle longevity and fecundity [17,21,28,29]. Under natural conditions, ladybeetles consume pollen and nectar simultaneously when foraging for flowers. Thus, in this study, we used fresh flowers to evaluate the effect of floral resources on *H. axyridis* longevity and fecundity instead of a single pollen diet.

In northern China, *Rhopalosiphum padi* and *Sitobion avenae* (Fabricius) the critical wheat aphids in winter wheat, and ladybeetles of *H. axyridis* and *P. japonica* are the most important natural enemies in wheat aphid biocontrol [5]. Furthermore, a laboratory experiment proved that *R. padi* is adequate prey and that *H. axyridis* larvae fed on *R. padi* can complete development and produce hatchable eggs. However, under the same laboratory conditions, compared with *R. padi*., *Acyrthosiphon pisum* Harris and *Schizaphis graminum* Rondani were the most suitable prey for *H. axyridis* [30]. Moreover, adult ladybeetles are more diffusible than larvae and often migrate to ingest supplementary food in non-crop habitats, and in our preliminary experiment, the *C. monnieri* flower can prolong larval and adult survival but did not fulfill larval development and adult reproduction of *H. axyridis*. These experiments addressed the following two questions: (1) Are *C.monnieri* flowers a suitable food for *Harmonia axyridis*? (2) Do the *C. monnieri* flowers enhance the development and reproduction performance of Harmonia axyridis? In this study, we measured the effect of nutritional supplements derived from *C. monnieri* fresh flowers on the development and reproduction of *H. axyridis* under wheat aphids-only diet conditions. First, the benefit of *C. monnieri* flowers on the development and reproduction of *H. axyridis* was compared with a wheat aphids-only diet. Second, the effects of *C. monnieri* as supplementary food on the longevity and fecundity of *H. axyridis* adults were assessed.

## 2. Results

### 2.1. Effects of Flower Resources on the Larval Development of H. axyridis

C. monnieri flowers significantly improved the larval survival rate of *H. axyridis*. When reared on a wheat aphids-only diet, only 77.3% of the larvae successfully reached adulthood, whereas the percentage was 90.7% for larvae offered a mixed diet of wheat aphids plus flowers (χ^2^ = 10.12, df = 1, *p* = 0.0015) (Figure 1). The larval developmental time was significantly shorter for H. axyridis when fed wheat aphids and flowers. The 1.8-day reduction in larval developmental time was accounted for by the third instar and fourth instar and the pupal period (Table 1) (Mann–Whitney U test, Z = −3.643, *p* < 0.001; Z = −5.901, *p* < 0.001; Z = −6.868, *p* < 0.001). There were no significant differences in first instars (Table 1) (Z = −0.020, *p* = 0.984) and second instars (Table 1) (Z = −0.593, *p* = 0.554) among treatments.

*C. monnieri* flowers yielded heavier pupal weights and newly emerged adult weights than aphids-only treatments. The pupal and adult weights of larvae reared on aphids plus flowers were significantly heavier than those of larvae reared on aphids-only (*t*-test: t = −5.745, df = 58, *p* < 0.001; *t*-test: t = 5.753, df = 58, *p* < 0.001) (Figure 2a,b). *H. axyridis* larvae fed on the aphid plus flowers diet had larger adults than those fed on aphids-only diets. Diet had a similar effect on adult body length for both females and males. The male and female body lengths on the mixed diet were significantly longer than the males raised from larvae reared on the prey-only diet (*t*-test: t = −2.372, df = 58, *p* = 0.021; *t*-test: t = −3.535, df = 58, *p* = 0.001) (Figure 2c,d). The sex ratios of adults obtained on aphids-only and aphid-plus flower diets averaged 1.19:1 and 1.23:1 (male: female), respectively.

### 2.2. Effects of Flower Resources on the Longevity and Reproduction of H. axyridis Adults

#### 2.2.1. Longevity

In Experiment I, flowers were continuously available in both larval and adult stages, and the longevity of male ladybeetles was significantly prolonged compared with the longevity in the aphids-only treatment. Male ladybeetles lived for 18.39 days longer with than without flowers (*t*-test: t = −3.309, df = 59, *p* = 0.002) (Figure 3b). The longevity of female ladybeetles was not significantly different between treatments (*t*-test: t = −1.837, df = 58, *p* = 0.071) (Figure 3a). 

In Experiment II, provided *C. monnieri* flowers as supplementary food to *H. axyridis* adults, the longevity of female ladybeetles and male ladybeetles was not significantly different between treatments (*t*-test: t = 0.646, df = 58, *p* = 0.521; *t*-test: t = 0.880, df = 58, *p* = 0.382) (Figure 4a,b).

#### 2.2.2. Reproduction

Reproduction was characterized using four parameters: the pre-oviposition period, the oviposition period, fecundity, and the hatching rate of eggs.

In Experiment I, flowers were continuously available during the larval and adult stages. Fecundity, estimated by the total number of eggs laid during the adult stage, was significantly increased by the presence of flowers. Females reared on the mixed diet laid a mean of 862.8 ± 63.77 whereas 532.73 ± 9.16 eggs were reared on the prey-only diet (*t*-test: t = −3.794, df = 58, *p* < 0.001) (Figure 3c). *C. monnieri* flowers increased egg production by 1.62 times. C. monnieri flowers also significantly prolonged the oviposition period by 9.2 days (*t*-test: t = −2.799, df = 58, *p* = 0.007) (Figure 3e). Furthermore, the egg-hatching rate was significantly improved on the mixed diet compared with the egg hatch rate on the aphid-only diet: over 67.46 ± 1.80% of H. axyridis eggs hatched on wheat aphids plus flowers versus 53.25 ± 3.45% (*t*-test: t = −3.658, df = 58, *p* = 0.001) (Figure 3f) on the aphid-only diet. There were no significant differences in the preoviposition period among treatments (nonparametric tests Mann–Whitney U test Z = −1, *p* = 0.317) (Figure 3d).

In Experiment II, flowers were only available in the adult stage, and the presence of flowers had a significant effect on the fecundity of *H. axyridis* and increased egg production by 1.41 times. Females reared on the aphids plus flowers diet laid a mean of 1027.13 ± 83.11 eggs with 731.37 ± 64.99 eggs reared on the aphids-only diet (*t*-test: t = −2.974, *df* = 58, *p* = 0.004) (Figure 4c). However, there were no significant differences in the preoviposition period (*Z* = −0.721, *p* = 0.471) (Figure 4d), the oviposition period (*t*-test: t = −0.852, *df* = 58, *p* = 0.398) (Figure 4e) or the egg hatching rate among treatments (*t*-test: t = 1.253, *df* = 58, *p* = 0.215) (Figure 4f).

## 3. Discussion

*H. axyridis* is a polyphagous predator that has been recognized as a biological control agent for several insect pests. *H. axyridis* ingests nectar, and pollen from flowering plants, and the floral resources are used by coccinellids to increase survival, reduce mortality, supply energy, and enhance reproductive capacity. In this study, we found that the flowering plants *C. monnieri* provide supplementary food that affects the development, longevity, and fecundity of *H. axyridis*. *H. axyridis* larvae completed development in both diets, with shorter developmental times, higher larval survival rates, heavier adult weights, larger body sizes, and more egg production resulting from the provision of *C. monnieri* flowers. 

### 3.1. Development

Larval performance was substantially enhanced when reared on the aphid plus flower diet compared to the aphids-only diet. Experiment I showed that *H. axyridis* larvae reared on the aphid plus flower diet had a significantly shorter larval duration and greater pupal weight and adult body length than the aphids-only treatment. This may indicate that the mixed diet of aphids and *C. monnieri* flowers is more suitable to *H. axyridis* larvae. J.P. Michaud wisely suggested that diets that support completed development are suitable diet, and the faster the development and/or the larger the adult the more suitable diet [31]. Similar to our study, many studies on coccinellids have found that the more suitable food resources typically yield shorter developmental times and heavier adults [32,33,34]. 

In this study, the essential nutrients provided by the pollen or nectar of *C. monnieri* flowers were beneficial to *H. axyridis*. Hodek and Honek suggested that many polyphagous coccinellids may have a mixed feeding habit, in which they select a favorable balance of important nutrients from various foods, including plant materials [35]. Flora resources, such as pollen and nectar, are excellent sources of essential insect nutrients [7,36]. Protein is one of the most abundant nutrients in pollen, typically comprising 12–61% of the dry weight [37]. Moreover, lipids, essential amino acids, fructose, glucose, and sucrose are present in most pollens [37]. There were some strong and positive effects on larval performance by mixing prey-only diets with pollen. The larvae of *H. axyridis* fed both flowers (*Fagopyrum esculentum* Moench, Polygonaceae) and prey (*Spodoptera littoralis* Boisduval caterpillars, Lepidoptera: Noctuidae) developed better than larvae fed each diet separately [38]. *Adalia bipunctata* (L.) (Coleoptera: Coccinellidae) fed on a mixed diet of frozen bee pollen and *Ephestia kuehniella* (Lepidoptera: Phycitidae) eggs had a faster development than on a diet of only *E. kuehniella* eggs [39]. Likewise, *Coleomegilla maculata* (DeGeer) developed better on a mixture of aphids and corn pollen than on aphids or pollen alone [40]. However, supplying a diet of *E. kuehniella* eggs with bee pollen only shortened the pre-oviposition period of *H. axyridis* but did not affect other developmental and reproductive parameters [29]. In conclusion, *C. monnieri* flowers may provide floral resources for *H. axyridis* larvae, which contain nutrients that are not abundant in wheat aphids. The nutrition of *C. monnieri* flowers helps to improve some essential foods to promote the development of *H. axyridis* larvae. 

### 3.2. Adult Longevity

Floral resources available only at the adult stage showed no significant differences in the longevity of either male or female ladybeetles among treatments. However, flower resources are constantly provided since the first instar significantly prolongs the longevity of male *H. axyridis*. This may indicate that the mixed diet showed high fitness for larval development and contributed to a heavier adult weight. This may allow more nutrients to be reserved in the adult body, and the nutrients reserved in the larvae may promote adult longevity. Similarly, body weight and adult longevity studies showed that heavier adults also had the longest lifespan [41]. The quality of food available during the larval stages could influence adult longevity and body size, and adult ladybeetles with larger body sizes may have more nutrient reserves to facilitate longevity than smaller adults [42,43]. The lifespan of *Lasioderma serricorne* (F.) (Coleoptera: Anobiidae Fleming) is positively correlated with the fresh weight of newly emerged adults [42]. Furthermore, nectar has a better effect on the longevity of predators than pollen. Many studies found that pollen could prolong predator longevity but was less effective than sugar solutions and flowers [7,44]. In conclusion, the *C. monnieri* flowers provide floral resources that cannot prolong the longevity of *H. axyridis* adults directly, but the longevity performance of *H. axyridis* could be improved through the nutrition reserved at the larval development stage.

### 3.3. Reproduction

The availability of *C. monnieri* flowers significantly enhanced the adult reproductive performance of *H. axyridis*. The fecundity of *H. axyridis* was significantly enhanced, regardless of whether *C. monnieri* flowers were provided at the adult stage or during the entire lifespan. In experiment II, females reared on an *R. padi*-only diet laid a mean of 731.37 eggs. Similar to our study, under laboratory conditions, *H. axyridis* fed on *R. padi* produced a mean of 791 eggs [30]. *C. monnieri* flowers provided at the entire lifespan of *H. axyridis* increased egg production 1.62 times, and those only provided at the adult stage increased egg production 1.44 times. Moreover, the number of oviposition days and egg hatchability were significantly higher in the aphid plus flower diet than in the aphids-only diet when flower resources were provided from the first instars of *H. axyridis*. This may indicate that *C. monnieri* flowers provide sufficient pollen for *H. axyridis*, and the protein of the pollen promoted the reproduction of *H. axyridis* adults. 

Reproduction requires food resources that are high in protein, and a mixture diet of prey and non-prey food usually supports greater reproduction output [7,16,45,46]. Riddick and Barbosa indicated the various pollen sources comprise adequate adult food for *Coleomegilla maculate fuscilabris* (Mulsant), even though a diet including animal protein usually results in higher fecundity [44]. Likewise, *A. bipunctata* (L.) fed a mixed diet of frozen bee pollen and *E. kuehniella* eggs had higher oviposition and egg hatching rates than *E. kuehniella* eggs only [39]. *Orius majusculus* (Anthocoridae: Orius) continued to lay eggs in the presence of alyssum species, but the maximum reproduction of *O. majusculus* was only possible under conditions of simultaneous availability of prey food [47]. A mixed diet of clover pollen with *E. kuehniella* eggs is a suitable diet for rearing *H. axyridis*, significantly increasing egg hatchability [17]. When pollen was added to a prey-only diet, a significantly larger proportion of ladybeetle eggs hatched over the following 3 days than during the previous 3 days [34]. In conclusion, the *C. monnieri* flowers significantly enhance the adult reproduction performance of *H. axyridis. H. axyridis* can compensate for a suboptimal diet of prey food by supplementary feeding on *C. monnieri* flowers. 

### 3.4. Implications for Conservation Biological Control

Understanding the mechanisms of floral resources on ladybeetle longevity and fecundity is essential to designing agroecosystems that support predator populations for pest control [48]. The present results indicated that *C. monnieri* flowers are a suitable supplementary food for promoting the development and reproduction of ladybeetles. Therefore, *C. monnieri* may be utilized as a candidate flowering plant to enhance the population of *H. axyridis* in crop systems where *H. axyridis* is the dominant natural enemy. Moreover, *C. monnieri* can attract predatory natural enemies, such as *Propylaea japonica*, *Hippodamia variegata*, *Chrysoperla sinica*, and *Episyrphus balteata*, especially during the flowering period [26]. Floral resources are also essential food for hoverflies and lacewing adults since they require nectar and pollen for survival and reproduction [49,50,51]. Future research should assess the fitness of *C. monnieri* flowers for other predators and evaluate the effects of *C. monnieri* plantings on pest suppression in a diversity of adjacent crops.

## 4. Materials and Methods

### 4.1. Experimental Insects and Plant Material

#### 4.1.1. Wheat Aphids *Rhopalosiphum padi* Linnaeus

*Rhopalosiphum padi* Linnaeus used in the experiments were collected from a wheat field in Jiyang, Shandong Province, China. Wheat aphids were reared on wheat seedlings as prey provided to *H. axyridis*.

#### 4.1.2. *Harmonia axyridis*

*H. axyridis* were reared under laboratory conditions in nylon mesh net cages (55 × 80 × 32 cm). They were fed ad libitum with *R. padi*, reared on wheat seedlings, and allowed to reproduce freely. The colony was maintained in an environmentally controlled chamber at 25 ± 2 °C, 65 ± 5% relative humidity (RH), and an L16:D8 h photoperiod. All pupae used in Experiment II were offspring of the third generation (the fourth generation). All newly hatched first instars of *H. axyridis* used in Experiment I were the offspring of the fourth (fifth) generation.

#### 4.1.3. *Cnidium monnieri* (L.) *Cusson*

*C. monnieri* was planted in Jiyang, Shandong Province. Every five days, fresh flowers were cut from the fields and preserved in a plastic bucket filled with water. For the flower plus aphid diet treatment, flowers were provided in bouquets and preserved in a plastic container. The bouquet consisted of 15 *C. monnieri* flowers with 12 cm stems wrapped with cotton at the bottom, placed into a 50 mL plastic container containing tap water. Before the experiments, the insects were carefully removed from the stems and flowers using a brush.

### 4.2. The Effect of C. monnieri Flowers on the Development and Reproduction of H. axyridis (Experiment I)

Two diets, aphids-only and aphid plus *C. monnieri* flowers, were tested. Each treatment started with 150 first-instar larvae, which were then left to develop into adults. For the experiment, newly hatched larvae of *H. axyridis* were randomly selected from egg clusters in the colony (the fifth generation) and carefully transferred by a soft brush to individual 300 mL plastic cups (diameter: 6.8 cm, height: 6.7 cm) with punctured lids for air exchange. Each cup contained a single larva, and water was provided by a moist cotton ball fitted into a 3.5 mm plastic dish. *R. padi* were offered ad libitum and refreshed daily. Each larva of *H. axyridis* reared on the aphid plus flower diet was provided with one flower, and the flowers were replaced every day. The immature stage was the period from the first instar to adult emergence. The developmental duration and survival of immature stages were monitored daily. The color patterns and presence of exuviae were used to identify the larval instars [3]. The experiment was conducted in an environmentally controlled chamber at 25 ± 2 °C, 65 ± 5% relative humidity (RH), and an L16:D8 h photoperiod. 

Pupae were weighed on the second day of pupation, and adult fresh body weight was measured during adult emergence. Pupae and newly emerged adults were weighed on a semi-microbalance Sartorius Genius SQP QUINTIX224-1CN (Sartorius AG, Göttingen, Germany) (±0.01 mg). The body length was measured at the time of death. Using an ocular micrometer (Nikon SMZ 745T, Tokyo, Japan), the distance between the anterior margin of the pronotum and the posterior margin of the elytra was measured.

After eclosion, adults were reared on wheat aphids 48 h before the experiment. The sex of the newly emerged adults was determined. Thirty pairs were formed from individuals who had experienced the same diet. They were subjected to the same diet as during immature development and were left to reproduce. Each pair was then placed in a plastic container (13.4 cm wide, 15.8 cm long, 18 cm high), and the top of the arena was covered with nylon mesh to allow ventilation. In each replicate, once the male died, a newly emerged male was placed in the container, and the experiment ended with the death of the female in each pair. During the experiment, wheat aphid was offered ad libitum and refreshed daily. Fresh flowers were replaced every five days, and water was provided by a moist cotton ball fitted into a 3.5 mm plastic dish and replaced every day. Females of *H. axyridis* were provided with small pieces of folded brown paper towels as a substrate for oviposition. The number of daily oviposition eggs was determined by checking the plastic container and folded brown paper towels daily. All eggs were collected daily and placed in separate plastic Petri dishes lined with moist absorbent paper at the base. The plastic Petri dish was labeled with the date, number of females, and number of eggs. The number of hatched eggs was determined by daily observations, and newly emerged larvae were counted and removed over time.

Reproduction was characterized using four parameters: pre-oviposition period, oviposition period, fecundity, and hatching rate of eggs [29]. The pre-oviposition and oviposition periods were determined by checking the Petri dishes daily for oviposited eggs. In addition, longevity and fecundity were assessed until the female died. Fecundity was determined by the total number of eggs laid by the females. The egg hatching rate was determined by the total number of eggs hatched from egg clusters as a proportion of the total number of eggs laid by the female during the day.

### 4.3. The Effect of C. monnieri Flowers on the Longevity and Reproduction of H. axyridis Adults (Experiment II)

To evaluate the fecundity and longevity of *H. axyridis* adults feeding on *C. monnieri* flowers and wheat aphids, two diets of aphids-only and aphid plus *C. monnieri* flowers were tested. The two diets comprised thirty replicates per diet, and each replication started with one pair of *H. axyridis* adults (48 h). Each pair of *H. axyridis* adults was placed in the same-sized plastic container as in Experiment I (the containers used at the adult stage). The experiment procedure and conditions were the same as in Experiment I.

### 4.4. Data Analysis

Data were checked for normality using the Kolmogorov–Smirnov (K-S) test. Larval survival curves were analyzed using the log-rank (Mantel-Cox) test in GraphPad Prism 6.0 (GraphPad, Inc., La Jolla, CA, USA). The data for the pupa weights, freshly emerged adult weights, adult body lengths, female and male longevity, fecundity, and hatching rate of eggs were normally distributed and analyzed using an independent-sample *t*-test to assess the treatment effect. Nonparametric tests (two independent samples, Mann–Whitney U tests) were performed on the developmental duration of the larva, pupa, the total immature period, and the preoviposition days. All statistical analyses were conducted using SPSS ver. 20 for Windows software package (IBM Corp., Armonk, NY, USA).

## 5. Conclusions

*C. monnieri* flower is a suitable supplementary food for *H. axyridis* larvae and adults and positively affects the survival, development, and reproduction performance of *H. axyridis*. *H. axyridis* larvae with shorter developmental times, higher larval survival rates, heavier adult weights, and larger body sizes resulting from the provision of *C. monnieri* flowers in the aphids-only diet. Regardless of whether the flowers were provided at both larval and adult stages or only at the adult stage, the fecundity of *H. axyridis* was significantly enhanced.

## Figures and Tables

**Figure 1 plants-12-01786-f001:**
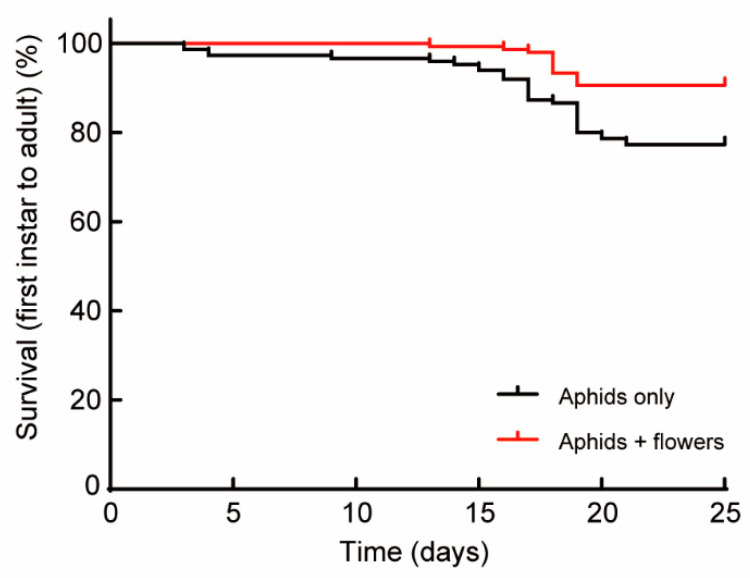
The survival rate of *Harmonia axyridis* larvae (from first instar to adult) reared on different diets. Survivorship curves are significantly different based on a log-rank (Mantel-Cox) test (*χ*^2^ = 10.12, *df* = 1; *p* = 0.0015).

**Figure 3 plants-12-01786-f003:**
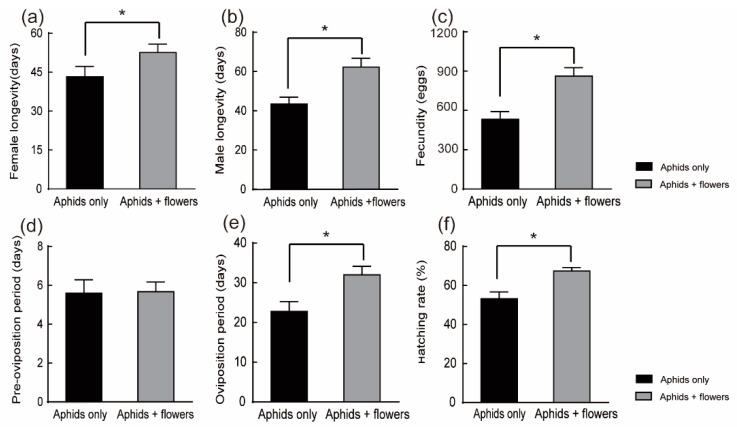
Experiment I: Effects of different diets on adult longevity and reproduction of *H. axyridis*. Flowers were continuously available in both larval and adult stages. (**a**) Female longevity. (**b**) Male longevity. (**c**) Fecundity. (**d**) Preoviposition period. (**e**) Oviposition period. (**f**) Hatching rate. Bars indicate the standard error (SE). * denote significant differences between the diets of feeding on aphids + flowers and only on aphids using a *t*-test (**a**–**c**,**e**,**f**) or Mann–Whitney U tests (**d**) (*p* < 0.05).

**Figure 4 plants-12-01786-f004:**
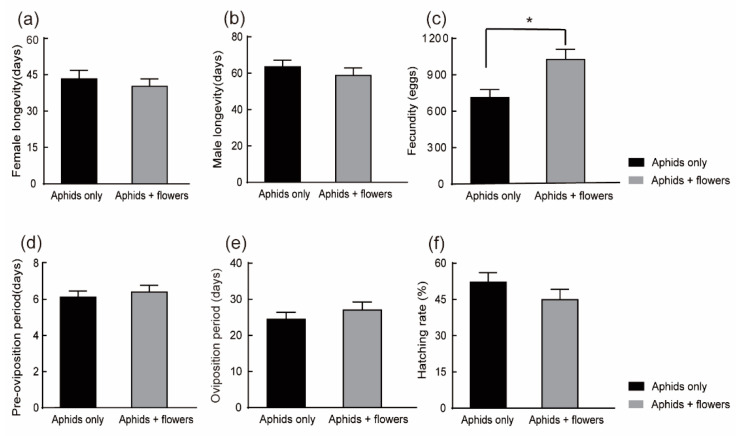
Experiment II: Effects of different diets on adult longevity and reproduction of *H. axyridis.* Flowers were only available at the adult stage. (**a**) Female longevity. (**b**) Male longevity. (**c**) Fecundity. (**d**) Preoviposition period. (**e**) Oviposition period. (**f**) Hatching rate. Bars indicate the standard error (SE). * denote significant differences between the diets of feeding on aphids + flowers and only on aphids using a *t*-test (**a**–**c**,**e**,**f**) or Mann–Whitney U tests (**d**) (*p* < 0.05).

**Figure 2 plants-12-01786-f002:**
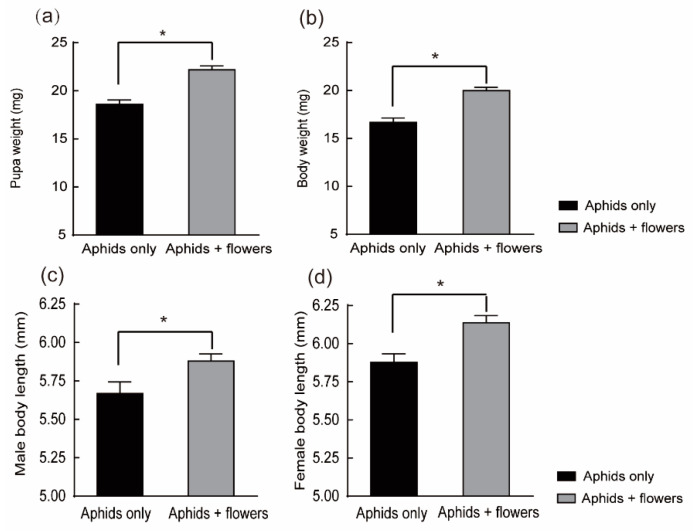
Effects of different diets on the pupa weight, adult weight, and adult body length of *H. axyridis.* (**a**) The pupal weight. (**b**) Fresh adult weight. (**c**) Male body length. (**d**) Female body length. * denotes significant differences between the diets of aphids + flowers and only aphids using a *t*-test (*p* < 0.05) (Experiment I).

**Table 1 plants-12-01786-t001:** Duration of larval instars, mean larval period, mean pupal period, and mean total developmental period of *H. axyridis* receiving different diets (all values in days).

Diet		Aphids Only (Days) (±SE)	Aphids + Flowers (Days) (±SE)
Larva	First instar	1.5086 ± 0.0473 A	1.5074 ± 0.0437 A
	Second instar	1.8190 ± 0.0427 A	1.8015 ± 0.0394 A
	Third instar	2.0948 ± 0.0531 A	1.8456 ± 0.0490 B
	Fourth instar	7.6207 ± 0.1342 A	6.4191 ± 0.1240 B
Larval period	First to fourth instar	13.0431 ± 0.1344 A	11.5735 ± 0.1241 B
Pupal period		4.3276 ± 0.0615 A	3.9853 ± 0.0568 B
Total developmental period		17.3707 ± 0.1283 A	15.5588 ± 0.1210 B

All means are followed by SE, and means followed by different letters indicate significant differences (Mann–Whitney U tests, *p* < 0.05).

## Data Availability

The datasets generated during and/or analyzed during the current study are not publicly available due to degree related but are available from the corresponding author on reasonable request.

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
