# Peer review of "Cnidium monnieri (L.) Cusson Flower as a Supplementary Food Promoting the Development and Reproduction of Ladybeetles Harmonia axyridis (Pallas) (Coleoptera: Coccinellidae)"

_plants, 2023, doi:10.3390/plants12091786_

Round 1

Reviewer 1 Report

Review

Review of the paper titled: Cnidium monnieri (L.) Cusson Flower as a Supplementary Food Promoting the Development and Reproduction of Ladybeetles Harmonia axyridis (Pallas) (Coleoptera: Coccinellidae). The work concerns the phenomenon of enriching the diet of predatory insects with plant substances. The work presents and documents well-conducted experiments and is, in my opinion, well-studied statistically. However, the presented results are nothing revealing, the authors themselves cite many works that describe similar dependencies. Therefore, the work should be treated as contributing.

The work is written in good language, although I, as a non-native speaker, do not make linguistic remarks.

Shortcomings that I see at work:

1. The research described is more about insects than plants, and therefore I don't think it fits the profile of the journal (but here the decision is up to the editor).

2. This type of research should relate to some general hypothesis, in this case, for example, a mixed diet increases insect performance. There is no such reference in the work. Further, the authors should clearly specify their research questions and hypotheses. It's not at work either.

3. The correct wording of the above-mentioned points makes it easier to write a discussion, which in this case is actually a repetition of the results chapter with references to some aspects of research by other authors. The discussion of the results should explain the observed phenomenon and explain the reasons. In this case, for example, what substances present in pollen increase the performance of insects.

All these things are fixable without the need for new research, just rethink and rewrite the discussion.

Detailed notes.

The figures are too small and therefore invisible.

The terms should be unified: R. padi only diet, aphid-only, aphids-only ,prey-only

Line 371-373 Two very similar sentences, perhaps one is unnecessary.

Author Response

Response to Reviewer 1 Comments

We would like to thank you for your careful reading, constructive comments and constructive suggestions, which has significantly improved the presentation of our manuscript. We have revised the manuscript to address the criticisms that the reviewers did note. The following is a point-to-point response to your comments.

Point 1: The research described is more about insects than plants, and therefore I don't think it fits the profile of the journal (but here the decision is up to the editor).

Response 1: We gratefully appreciate for your valuable comment. We totally understand your concern. In our study, we focused on the effects of nutrients provided by plants on the growth and development of natural enemy Harmonia axyridis. The improvement of plant supplementary food on the natural enemy population may improve their biocontrol capacity of crop insect pest. Thus, topic of this manuscript meets the “plant–insect interaction” subject of the special issue.

Point 2: This type of research should relate to some general hypothesis, in this case, for example, a mixed diet increases insect performance. There is no such reference in the work. Further, the authors should clearly specify their research questions and hypotheses. It's not at work either.

Response 2: Thank you for pointing out the details. We have been specified our research questions in introduction (line125-127) of the manuscript.

Point 3: The correct wording of the above-mentioned points makes it easier to write a discussion, which in this case is actually a repetition of the results chapter with references to some aspects of research by other authors. The discussion of the results should explain the observed phenomenon and explain the reasons. In this case, for example, what substances present in pollen increase the performance of insects.

Response 3: Thank you for the above suggestion. In the revised manuscript, we have re-written the discussion.

Point 4: The figures are too small and therefore invisible.

Response 4: We are very sorry for the figures size in this manuscript caused inconvenience in your reading. We have resized the figures in the manuscript. Thank you for pointing out this problem in our manuscript.

Point 5: The terms should be unified: R. padi only diet, aphid-only, aphids-only, prey-only

Response 5: Thank you for pointing out this problem in our manuscript. The word has been changed in the revised version. Change “Aphid only” and “prey only” to “aphids-only”.

Point 6: Line 371-373 Two very similar sentences, perhaps one is unnecessary.

Response 6: Thank you for pointing out this problem in our manuscript. We have cut out the repeated sentences.

Reviewer 2 Report

I recommend to accept this paper after minor revision. I have added some comments in the pdf file.

Author Response

Response to Reviewer 2 Comments

We would like to thank you for your careful reading, positive comments and constructive suggestions, which has significantly improved the presentation of our manuscript.

We have carefully considered all comments from you and revised our manuscript accordingly. The manuscript has been double-checked, the format error in Latin names have been corrected, and the typos errors has been corrected, and a conclusion chapter has been added to the manuscript. Many thanks for your kind help!

Round 2

Reviewer 1 Report

I have no comments.

Author Response

We would like to thank you for your careful reading and your positive comments. Many thanks for your kind help.